# Interaction between central obesity and frailty on the clinical outcome of peritoneal dialysis patients

**Gordon Chun-Kau Chan**[☯]**, Jack Kit-Chung N. G., Kai-Ming Chow, Vickie Wai-Ki Kwong, Wing-Fai Pang, Phyllis Mei-Shan Cheng, Man-Ching Law, Chi-Bon Leung, Philip Kam-Tao L. I., Cheuk Chun Szeto**[ID]*[☯]

Department of Medicine & Therapeutics, Carol & Richard Yu Peritoneal Dialysis Research Centre, The Chinese University of Hong Kong, Shatin, Hong Kong, China

☯ These authors contributed equally to this work.
* ccszeto@cuhk.edu.hk

**Data Availability Statement:** All relevant data are within the paper.

**Funding:** This study was supported by the Chinese University of Hong Kong (CUHK) research

## Abstract

### Background

Frailty and obesity contribute to the adverse clinical outcome of peritoneal dialysis (PD) patients, but the interaction between frailty and obesity remains uncertain.

### Objective

To examine the interaction between frailty and obesity on the clinical outcome of PD patients.

### Design

Single centre prospective observational cohort study.

### Patients

267 prevalent Chinese PD patients were recruited.

### Measurements

Frailty was identified by a standard score. General and central obesity were determined by body mass index (BMI) and waist-hip ratio (WHR), respectively. Body composition was assessed by bioimpedance spectroscopy. All patients were followed for two years. Outcome measures included all-cause as well as cardiovascular mortality and hospitalization.

### Results

Of the 267 patients, 120 (44.9%) were frail. Frail individuals were more likely to have central obesity (p < 0.001) but not general obesity. Although WHR did not predict patient survival, there was a significant interaction between WHR and frailty on patient survival and cardio-vascular survival (p = 0.002 and p = 0.038, respectively). For patients without frailty, the two-year cardiovascular survival was 91.3% and 74.4% for those with and without central

accounts 6901031 and 7101215. The funders had no role in study design, data collection and analysis, decision to publish, or preparation of the manuscript.

**Competing interests:** The authors declare no conflict of interest.

obesity, respectively (p = 0.002). For patients with frailty, however, the two-year cardiovascular survival was 64.6% and 66.7% for those with and without central obesity, respectively (p = 0.6). For patients without frailty, the number of hospital admission for cardiovascular disease over 2 years were 0.12 ± 0.37 and 0.34 ± 0.72 for those with and without central obesity, respectively (p = 0.03). For frail patients, however, the number of hospital admission was similar between those with and without central obesity.

## Conclusion

There is a significant interaction between frailty and central obesity on the outcome of PD patients. The protective role of central obesity is only apparent in PD patients without frailty but not the frail ones, and there is a little prognostic value of general (non-central) obesity.

## Introduction

### Background

Frailty is described by Fried et al. [1] as a clinical syndrome involving at least three of the following features: unintentional weight loss (10 lbs in one year), self-reported exhaustion, weakness (as determined by grip strength), slow walking speed, and low physical activity. It is highly prevalent among patients on dialysis, with a prevalence of up to 67.7% in previous report [2].

Physical frailty has important health consequences. Frail patients are more susceptible to the development of adverse outcomes when faced with a stressor. It also leads to a decline in physical function, gait disability, increase in the risks of fall. As a result, frail peritoneal dialysis (PD) patients have a higher hospitalization risk and their hospital stay tends to be prolonged [3]. Physical frailty is also a well-established independent predictor of adverse outcomes including mortality in pre-dialysis end stage renal disease patients [4], hemodialysis (HD) patients and PD patients [3, 5].

Physical frailty causes alteration in body composition. Frail patients tend to suffer from obesity [6–9]. On the other hand, PD patients are more susceptible to central obesity [10] due to exposure to glucose-containing dialysate fluid. In contrast to general obesity, central obesity confers a worse survival in dialysis patients [11]. Previous study has shown synergistic effect on mortality between physical frailty and central obesity in HD patients. However, the interaction between physical frailty and body composition has not been explored.

### Objectives

The goal of this study of was to dissect the internal relationship between obesity and frailty on the outcome of PD patients.

## Methods

### Study design

This is a single centre prospective cohort study. The study was approved by the Joint Chinese University of Hong Kong—New Territories East Cluster Clinical Research Ethics Committee. All study procedures were in compliance with the Declaration of Helsinki. All patients provided a written informed consent before enrollment in the study.

## Study population

We recruited 267 prevalent PD patients from a single dialysis unit from 1 January 2015 to 31 December 2016. Patients with expected survival of less than 3 months, or those who were planned to receive kidney transplantation in 3 months, were excluded.

After written informed consent, assessments of frailty, anthropometry, body composition and pulse wave velocity were measured at baseline. Other clinical and laboratory data were obtained by chart review. Clinical data comprises of patient's age, gender, body weight, height, primary diagnosis of renal disease, concomitant chronic medical illnesses including diabetes mellitus, ischemic heart disease, cerebrovascular accident, peripheral vascular disease, chronic hepatitis B and C infection, chronic lung disease, malignancy and immunological diseases. Laboratory assessment included serum albumin level, serum cholesterol (including total cholesterol, high density cholesterol, low density cholesterol) level, dialysis adequacy by Kt/V, residual renal function by measuring the residual glomerular filtration rate from urine. The Charlson Comorbidity Index (CCI) was used to assess the comorbidity load. The Comprehensive Malnutrition Inflammation Score (MIS) [12] and Subjective Global Assessment (SGA) [13] were used to assess nutritional state.

## Assessment of frailty

We used a validated Chinese questionnaire that consisted of 30 yes/no questions [3]. The questions involve assessment of subjective assessment of personal health, psychological state, physical state in terms of number of hospital or doctor visit and medication needs to be taken, body weight, need of assistance in different aspects of daily living and mobility. A total score was calculated; physical frailty was defined as a score of 6 or above [3]. To ensure patient adequate understanding on the 30 questions, all the recruited subjects were directly interviewed by designated well-trained interviewers at baseline.

## Anthropometric and body composition assessment

Anthropometric data were measured at recruitment. Body mass index (BMI) was used to assess general adiposity; general obesity was defined as BMI more than or equals 30 kg/m$^2$ respectively according to the definition from World Health Organisation (WHO). Waist hip ratio (WHR), as defined by the ratio between waist circumference to hip circumference, was used to assess central adiposity. Waist circumference was measured at the approximate midpoint between the lower margin of the last palpable rib and the top of the iliac crest, while the hip circumference was measured around the widest portion of the buttocks. Central obesity was defined as WHR more than or equals sex-specific sample median (females 0.92, males 0.98).

We used the multi-frequency bioimpedance spectroscopy device (Body Composition Monitor [BCM], Fresenius Medical Care, Germany), which was validated in dialysis patients [14, 15]. Briefly, electrodes were attached to one hand and one foot with the patient in a supine position. The following parameters were computed: extracellular water (ECW), lean tissue mass (LTM), adipose tissue mass (ATM), and volume of overhydration (OH).

Anthropometric and body composition assessments were performed with empty abdomen (before infusion of dialysate) by the same trained research nurse with at the time of recruitment.

## Pulse wave velocity

Pulse wave velocity (PWV) is a simple, non-invasive technique of assessing arterial stiffness. High levels of PWV analysis are indicative of increased rigidity and low distensibility of

vascular walls, along with poor vascular function [16]. PWV was measured by Vicorder device (SMT Medical GmbH&Co.), an automatic computerised recorder. The results were analyzed by the Complior Analyse program (Artech Medical, Pantin, France). Pressure-sensitive transducers will be placed over neck (carotid artery), wrist (radial artery), and groin (femoral artery), with the patient in the supine position. PWV of the carotid-femoral (CF-PWV) and carotid-radial (CR-PWV) territories will be calculated by dividing the distance between the sensors by the time corresponding to the period separating the start of the rising phase of the carotid pulse wave and that of the femoral and also the radial pulse waves. Measurements were performed at the beginning of study, and 1 year after recruitment.

### Outcome measures

All patients were followed for 2 years. Primary outcome measures were all-cause mortality and cardiovascular mortality. Secondary outcome measures were number of all-cause hospital admission, number of cardiovascular event-related hospitalization, and total length of hospital stay within study period. Cardiovascular mortality is defined by death due to peripheral vascular disease, myocardial infarction, heart failure, stroke and sudden cardiac death. In this analysis, receiving kidney transplantation, transferal to hemodialysis, transferal to another center and death from non-cardiovascular causes were considered as competing events.

### Statistical analysis

Statistical analysis was performed by SPSS for Windows software version 24 (SPSS Inc., Chicago). Descriptive data were presented as mean ± SD if normally distributed, or median (inter-quartile range) otherwise. Patients were grouped according to presence of physical frailty as defined above for analysis. Clinical parameters were compared by Student's t-test or one way analysis of variance (ANOVA) for continuous variables, and Chi-square test for categorical variables. P<0.05 was considered to be statistically significant in this study. All probabilities were two-tailed.

The number and duration of hospitalization were compared between non-frail and frail individuals. Age, dialysis vintage, Charlson Comorbidity Index, serum albumin, total Kt/V, normalised protein nitrogen appearance (NPNA) and residual glomerular filtration rate (GFR), which were potential confounding factors shown in previous studies [3, 17, 18], were added to linear regression model for adjustment. Backward stepwise elimination was used to determine the independent predictor of hospitalization.

Kaplan Meier plots were constructed for patient and cardiovascular survival rates. Log-rank test was used to compare between survival curves. Receiving kidney transplantation, transferal to hemodialysis, and transferal to another center were considered as competing events. Multivariate Cox regression models were constructed to further identify independent predictors of survival after adjustment of potential confounders with P values below 0.1 with univariate analysis. Backward stepwise analysis was used to remove insignificant variables. P<0.05 was considered to be statistically significant in this study. All probabilities were two-tailed.

## Results

### Baseline demographic characteristics

A total of 267 PD patients were recruited. Their baseline clinical and demographic data are summarized in Table 1 and S1 Table. Among them, 120 patients (44.9%) were physically frail (Fig 1). As compared to those without frailty, frail PD patients were older, more often required helper assistance in performing PD, had higher CCI, and worse nutritional state. Frail PD

**Table 1. Baseline clinical and demographic data.**

| | Not frail | Frail |
|---|---|---|
| No. of patients | 147 | 120 |
| Age (year) | 61.1 ± 11.8 | 64.2 ± 12.2 |
| Sex (M:F) | 75: 72 | 56: 64 |
| Duration of dialysis (months) | 44.3 ± 46.9 | 51.8 ± 57.4 |
| Blood pressure (mmHg) | | |
| Systolic | 142.8 ± 18.9 | 142.6 ± 20.9 |
| Diastolic | 75.8 ± 12.4 | 73.4 ± 12.7 |
| Renal Diagnosis, no of cases. (%) | | |
| Glomerulonephritis | 51 (34.7%) | 33 (27.5%) |
| Diabetic nephropathy | 42 (28.6%) | 56 (46.7%) |
| HTN | 18 (12.2%) | 12 (10%) |
| Polycystic kidney | 5 (3.4%) | 3 (2.5%) |
| Urological disease | 9 (6.1%) | 2 (1.7%) |
| Other / unknown | 22 (15%) | 14 (11.6%) |
| Comorbid disease, no of cases (%) | | |
| DM | 55 (37.4%) | 66 (55.0%) |
| IHD | 19 (12.9%) | 19 (15.8%) |
| CVA | 16 (10.9%) | 31 (25.8%) |
| Charlson Comorbidity Index | 5.1 ± 2.2 | 5.6 ± 2.2 |
| Nutritional Status | | |
| MIS | 6.57 ± 3.13 | 9.00 ± 3.46 |
| SGA | 5.38 ± 0.83 | 4.84 ± 0.92 |
| Total weekly Kt/V | 1.88 ± 0.49 | 1.76 ± 0.40 |
| Residual GFR (ml/min/1.73m2) | 1.70 ± 2.02 | 1.04 ± 1.74 |
| NPNA (g/kg/day) | 1.07 ± 0.25 | 1.03 ± 0.30 |
| Biochemical parameters | | |
| Hemoglobin (g/dL) | 9.72 ± 1.31 | 9.25 ± 1.26 |
| Albumin (g/L) | 34.10 ± 4.15 | 33.46 ± 4.16 |
| Pulse Wave Velocity (m/sec) | | |
| CF-PWV | 10.93 ± 2.38 | 11.19 ± 2.40 |
| CR-PWV | 10.34 ± 1.75 | 10.48 ± 1.80 |
| Peritoneal Transport State- 4-hour dialysate/plasma creatinine | 0.64 ± 0.13 | 0.65 ± 0.12 |
| Helper-assisted PD, no of cases (%) | 8 (5.4%) | 24 (20.0%) |
| Type of peritoneal dialysis, no of cases (%) | | |
| CAPD | 127 (86.4%) | 106 (88.3%) |
| CCPD | 4 (2.7%) | 2 (1.7%) |
| NIPD | 16 (10.9%) | 12 (10.0%) |
| Icodextrin dialysate use, no of cases (%) | 39 (26.5%) | 34 (28.3%) |
| Concomitant medications, no of cases (%) | | |
| Aspirin | 35 (23.8%) | 42 (35.0%) |
| Beta blocker | 99 (67.3%) | 72 (60.0%) |
| RAS blocking agent | 79 (53.7%) | 77 (64.2%) |
| Calcium channel blocker | 125 (85.0%) | 90 (75.0%) |
| Statin | 64 (43.5%) | 62 (51.7%) |
| Calcium containing phosphate binder | 115 (78.2%) | 84 (70.0%) |
| Vitamin D supplement | 64 (43.5%) | 62 (51.7%) |

(*Continued*)

**Table 1.** (Continued)

| | Not frail | Frail |
|---|---|---|
| Erythropoiesis-stimulating agents | 112 (76.2%) | 82 (68.3%) |

Data are expressed as mean ± standard deviation.

HTN, hypertensive nephrosclerosis; DM, diabetes mellitus; IHD, ischemic heart disease; CVA, cerebrovascular accident; MIS, Malnutrition Inflammation Score; SGA, Subjective Global Assessment; GFR, glomerular filtration rate; NPNA, normalized protein nitrogen appearance; CF-PWV, carotid-femoral pulse wave velocity; CR-PWV, carotid-radial pulse wave velocity; PD, peritoneal dialysis; CAPD, continuous ambulatory peritoneal dialysis; CCPD, continuous cycler peritoneal dialysis; NIPD, nocturnal intermittent peritoneal dialysis; RAS blocking agent, renin angiotensin system-blocking agent.

patients were also more likely to be diabetic, had lower residual renal function and hemoglobin level.

The body composition of patients with and without frailty are summarized and compared in Table 2. Frail patients had significantly higher WHR, and marginally higher BMI, than those without frailty, although the latter was not statistically significant. Among frail patients, the prevalence of central obesity and general obesity was 60.0% and 13.3% respectively, whereas among non-frail patients, the prevalence was 46.9% and 10.2% respectively. Frail patients had marginally higher adipose tissue mass than non-frail patients by bioimpedance spectroscopy, although the difference was not statistically significant. Frail patients also had significantly a higher overhydration volume than those without frailty, but their lean tissue mass was similar.

There was no significant difference in baseline CF-PWV and CR-PWV between patients with and without frailty. After 1 year of PD, CF-PWV increased from 11.04 ± 2.39 to 12.19 ± 3.17 m/sec (paired Student's t test, p = 0.003), whereas there was no significant change in CR-PWV (from 10.40 ± 1.77 to 10.73 ± 2.14 m/sec, paired Student's t test, p = 0.8). The baseline frailty score did not correlate with the change in CF-PWV (r = 0.145, p = 0.3) or CR-PWV (r = 0.081, p = 0.6) over one year. In central obese patients, there was a modest correlation between physical frailty and the change in CF-PWV although it did not reach statistical significance (r = 0.359. p = 0.08). There was no correlation between physical frailty and change in CR-PWV (r = 0.001. p = 0.9).

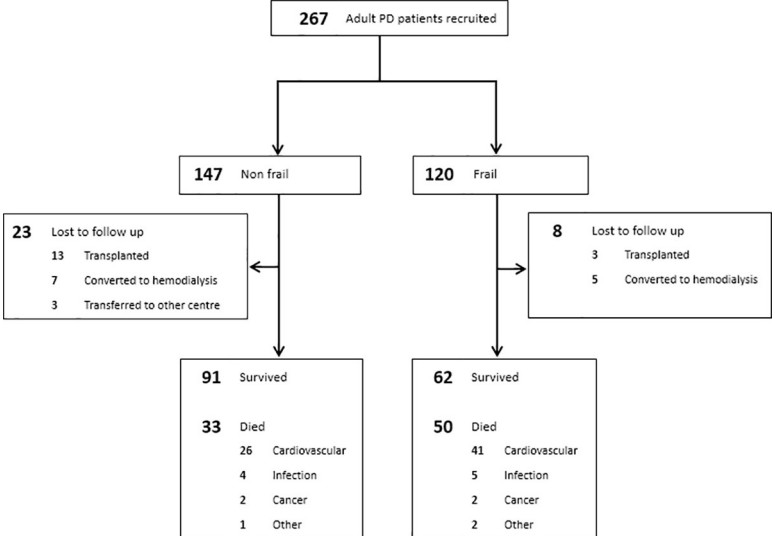

**Fig 1. Flow diagram of subjects' inclusion and their outcomes.**

**Table 2. Body anthropometry and composition by bioimpedance spectroscopy.**

|  | Not frail | Frail | P value |
|---|---|---|---|
| No. of patients | 147 | 120 |  |
| Body weight (kg) | 64.2 ± 12.9 | 66.3 ± 12.5 | p = 0.2 |
| Body height (cm) | 161.0 ± 7.5 | 160.0 ± 8.3 | p = 0.2 |
| Body mass index (kg/m$^2$) | 25.7 ± 4.1 | 26.0 ± 4.3 | p = 0.07 |
| Waist hip ratio (WHR) | 0.95 ± 0.07 | 0.99 ± 0.11 | p < 0.001 |
| Overhydration (OH) |  |  |  |
| Volume (L) | 2.90 ± 3.09 | 3.62 ± 2.99 | p = 0.08 |
| >1 L, no. of case (%) | 101 (72.7%) | 96 (87.3%) | p = 0.005 |
| Adipose tissue mass (kg) | 21.1 ± 11.6 | 23.1 ± 11.0 | p = 0.16 |
| Lean tissue mass (kg) | 38.0 ± 9.66 | 36.7 ± 9.67 | p = 0.3 |

## Patient survival

During the study period, 83 patients (31.1%) died. During this period, 16 patients (6.0%) underwent kidney transplantation, 12 (4.5%) were switched to chronic hemodialysis, and 3 (1.1%) were transferred to other renal center. The 2-year overall survival rate was 58.3% in frail and 77.6% in non-frail patients (log-rank test, p = 0.001). The 2-year cardiovascular survival was 65.8% in frail and 82.3% in non-frail individuals (log-rank test, p < 0.001). After adjusting for potential confounding factors by multivariate Cox regression analysis (Table 3), physical frailty remained a significant predictor for all-cause mortality and cardiovascular mortality. In these models, other significant predictive factors of all-cause mortality included age, CCI and serum albumin level. For cardiovascular mortality, CCI was the only other independent predictor.

## Interaction with WHR on survival

Neither WHR nor BMI predicted overall survival or cardiovascular survival. However, there was a significant interaction between central obesity and frailty on patient survival and

**Table 3. Univariate and multivariate Cox regression analysis.**

(A) Patient survival

|  | Univariate analysis | | Multivariate analysis | |
|---|---|---|---|---|
| Factors | AHR (95% CI) | P value | AHR (95% CI) | P value |
| Physical frailty | 2.126 (1.369–3.330) | p = 0.001 | 1.793 (1.094–2.939) | p = 0.021 |
| CCI | 1.350 (1.246–1.464) | p < 0.001 | 1.131 (1.015–1.259) | p = 0.025 |
| Age | 1.067 (1.043–1.091) | p < 0.001 | 1.069 (1.034–1.104) | p < 0.001 |
| Albumin | 0.881 (0.837–0.927) | p < 0.001 | 0.915 (0.859–0.975) | p = 0.006 |
| CF-PWV | 1.172 (1.070–1.284) | p = 0.001 |  |  |
| Overhydration | 1.085 (1.019–1.156) | p = 0.011 |  |  |

(B) Cardiovascular survival

|  | Univariate analysis | | Multivariate analysis | |
|---|---|---|---|---|
| Factors | AHR (95% CI) | P value | AHR (95% CI) | P value |
| Physical frailty | 2.959 (1.277–6.860) | p = 0.011 | 2.652 (1.134–6.200) | p = 0.024 |
| CCI | 1.282 (1.111–1.479) | p = 0.001 | 1.294 (1.109–1.509) | p = 0.001 |
| Age | 1.052 (1.011–1.095) | p = 0.012 |  |  |
| Albumin | 0.921 (0.839–1.011) | p = 0.085 |  |  |

CCI, Charlson Comorbidity Index; CF-PWV, carotid-femoral pulse wave velocity; BMI, body mass index; AHR, adjusted hazard ratio; CI, confidence interval; CV, cardiovascular.

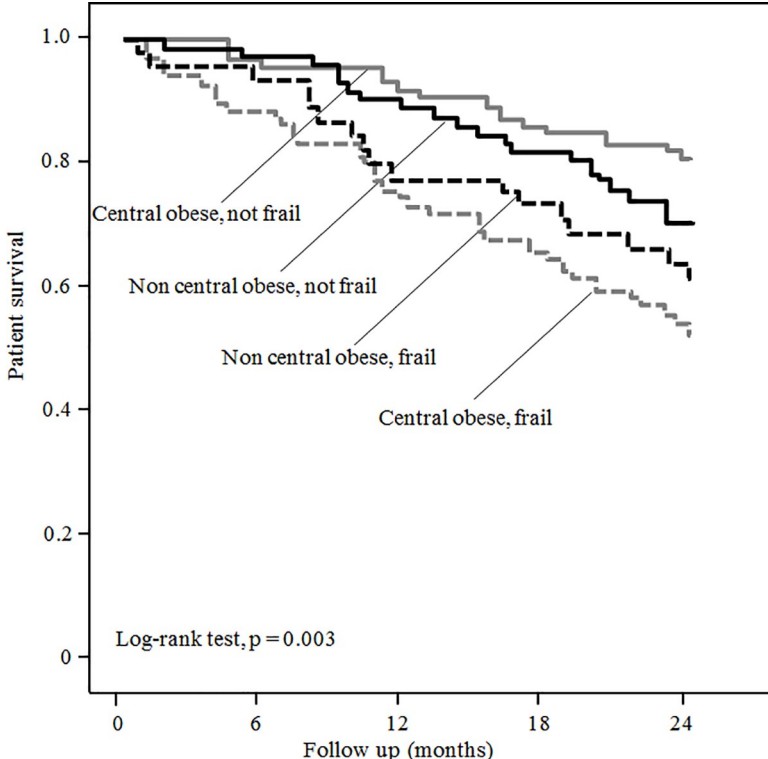

**Fig 2. Kaplan-Meier plot of patient survival according to the frailty state and central obesity.**

cardiovascular survival (p = 0.002 and p = 0.038, respectively, for interaction), while the interaction between general obesity and frailty on survival did not reach statistical significance (overall survival, p = 0.057 and cardiovascular survival, p = 0.156, respectively, for interaction). The 2-year patient survival of patients with central obesity was marginally lower than those without central obesity for the frail group (54.2% vs 64.6%, p = 0.3), but marginally higher than those without central obesity for the non-frail group (81.2% vs 74.4%, p = 0.2), although neither of the difference was statistically significant (Fig 2). Similarly, the 2-year cardiovascular survival of patients with central obesity was similar to those without central obesity for the frail group (64.6% vs 66.7%, p = 0.6), but significantly higher than those without central obesity for the non-frail group (91.3% vs 74.4%, p = 0.002).

Further subgroup analysis on 2-year patient survival was performed for patients on short duration of peritoneal dialysis (less than 1 year use). In short, the survival rate was higher among non-frail individuals, which is in line with the results of our whole cohort. Non-frail, central obese individuals had the best survival rate while frail, central obese individuals had the worst survival rate. The survival rates decreased in a stepwise manner, from 100% for non-frail central obese, 82.4% for non-frail non-central obese, 76.9% for frail non-central obese, to 61.5% for non-frail non-central obese (S1 Fig) though the difference did not reach significance (log-rank test, p = 0.069).

## Hospitalization

After 2 years of follow up, there were in total 984 episodes of hospital admission for a total of 10,695 days in our cohort; 45 patients (16.8%) did not require any hospitalization during the study period (Table 4). Among all, 88 episodes of hospitalization admission for a total of 1,190

**Table 4. Relation between physical frailty and hospitalization.**

|  | Not frail | Frail | P value |
|---|---|---|---|
| Number of patients | 147 | 120 |  |
| Without hospitalization, no. of patients (%) | 24 (16.3%) | 21 (17.5%) | p = 0.799[a] |
| Number of hospitalization |  |  |  |
| All-cause | 3.13 ± 3.51 | 4.37 ± 4.27 | p = 0.025[b] |
| Cardiovascular event | 0.24 ± 0.59 | 0.44 ± 0.92 | p = 0.049[b] |
| Total duration of hospitalization (days) | 31.05 ± 42.48 | 51.08 ± 68.99 | p = 0.068[b] |

Data are compared by

[a]Chi-square test and

[b]Mann Whitney U test.

days had cardiovascular events as the primary diagnosis. In short, frailty score had modest correlations with the number of hospitalization for all-cause (r = 0.158, p = 0.025) or cardiovascular event (r = 0.133, p = 0.049), total duration of hospitalization (r = 0.176, p = 0.068), although the last correlation did not reach statistical significance. In contrast, WHR did not correlate with number of hospitalization for all-cause (p = 0.4) or cardiovascular-event (p = 0.7), or the duration of hospitalization (p = 0.4). By using linear regression model after log transformation to adjust for potential confounders, frailty score remained an independent factor associated with the number of hospital admission for all-cause and total duration of hospitalization (Table 5).

We further performed subgroup analysis on hospitalization data according to physical frailty. In short, general obese patients were hospitalized more frequently with a longer hospitalization period, while central obese patients were hospitalized less frequently with a shorter period (Table 6). In non-frail patients, those with general obesity had more than a doubled number and duration of cardiovascular-event related hospitalization as compared to those without general obesity, while those with central obesity had halved number and duration of cardiovascular-event related hospitalization as compared to those without central obesity. A similar trend was also observed in frail patients, but the difference was not significant (S2 Table).

**Table 5. Multivariate linear regression analysis of hospitalization after log transformation.**

| (A) Number of hospitalization | | | |
|---|---|---|---|
| Factors | Unstandardized B | 95% CI | P value |
| Physical frailty | 0.998 | 0.024–1.971 | p = 0.045 |
| CCI | 0.372 | 0.101–0.644 | p = 0.007 |
| GFR | -0.348 | -0.602 – -0.094 | p = 0.007 |
| (B) Number of cardiovascular event-related hospitalization | | | |
| Factors | Unstandardized B | 95% CI | P value |
| CCI | 0.060 | 0.017–0.104 | p = 0.007 |
| Kt/V | -0.191 | -0.355 – -0.027 | p = 0.022 |
| (C) Total duration of hospitalization | | | |
| Factors | Unstandardized B | 95% CI | P value |
| Physical frailty | 14.295 | 0.032–28.558 | p = 0.049 |
| CCI | 4.090 | 0.954–7.227 | p = 0.011 |
| Kt/V | -14.483 | -26.696 – -2.270 | p = 0.020 |

CI, confidence interval; CCI, Charlson Comorbidity Index; GFR, glomerular filtration rate.

**Table 6. Relation between hospitalization and general versus central obesity in non-frail patients.**

|  | No central obesity | Central obesity | P value* | No general obesity | General obesity | P value* |
|---|---|---|---|---|---|---|
| Number of patients | 78 | 69 |  | 132 | 15 |  |
| Number of hospitalization |  |  |  |  |  |  |
| All-cause | 3.37 ± 3.94 | 2.86 ± 2.95 | p = 0.5 | 3.08 ± 3.59 | 3.60 ± 2.75 | p = 0.189 |
| Cardiovascular event | 0.34 ± 0.72 | 0.12 ± 0.37 | p = 0.030 | 0.20 ± 0.56 | 0.60 ± 0.74 | p = 0.002 |
| Duration of hospitalization (days) |  |  |  |  |  |  |
| All-cause | 35.22 ± 51.53 | 26.34 ± 28.70 | p = 0.9 | 29.67 ± 41.73 | 43.27 ± 48.39 | p = 0.076 |
| Cardiovascular event | 3.54 ± 9.16 | 1.45 ± 5.38 | p = 0.040 | 1.92 ± 7.04 | 8.13 ± 10.62 | p = 0.001 |

*Data are compared by Mann Whitney U test.

## Discussion

The present study identified frailty is prevalent among PD patients, and it is an independent predictor of adverse outcomes including a higher hospitalization rate, a longer duration of hospitalization, as well as a worse survival with higher all-cause and cardiovascular mortality rates. Our results also showed presence of significant interactions between central obesity and frailty in terms of hospitalization and survival in PD patients.

Physical frailty is traditionally regarded as a geriatric disease as it primarily affects elderly patients. As predicted, frail individuals are generally older in our cohort as shown in Table 1. When compared to HD, performing PD is relatively simpler and it does not require as much technical support as in HD. PD also does not require vascular access creation, and it offers a better cardiovascular stability [19–21]. PD is therefore commonly regarded as a better dialysis option over HD in elderlies. More than 30% to 75% of elderlies preferred PD over HD in different countries [22, 23]. Assisted PD is particularly more preferred to hemodialysis as assisted PD provides a better quality of life to frail elderly on dialysis [24–26]. Therefore, it is important to recognize and quantify the impact of physical frailty in PD patients. In fact, routine frailty screening has been suggested to be part of assessment for and maintenance care on PD [27]. However, most of the currently available literature in the field of physical frailty in dialysis population involved either HD [28] or a mixture of HD and PD cases [29]. This evidence are unable to reflect the burden of physical frailty in PD population, and cannot be applied to certain countries like Hong Kong, Taiwan, Thailand, Vietnam, as well as New Zealand and Australia, which have a relatively higher PD to HD case ratio compared to other western countries like the USA and Canada [30].

Patients with chronic kidney disease have a different body composition compared with the general population [31, 32]. The causes are multifold. Firstly, uremia, defects in homeostasis and metabolic derangements in chronic kidney disease cause anorexia, which results in malnutrition and protein energy wasting [33]. Secondly, dialysis treatment also induces protein loss and muscle catabolism [34, 35]. In particular, PD causes central obesity [10, 36] due to the exposure to glucose-containing dialysate fluid. On the other hand, physical frailty itself is also associated with similar change in body composition i.e. development of obesity with a reduction in muscle mass [7–9]. The overall combined effect on body composition by renal failure, dialysis treatment and physical frailty in PD patients were uncertain. Our study demonstrated a comparable total adipose tissue mass between frail and non-frail PD patients. It is worth to note that despite the total adipose tissue was similar between two groups, when comparing between general and central obesity, frail individuals were more central obese but not general obese. This observation highlights frail individual had a different fat composition, with a relatively higher visceral fat to subcutaneous fat ratio when compared to non-frail PD patients.

This may be the potential cause for an inferior survival in frail PD patients, which will be further elaborated.

In contrast to general obesity, central obesity correlates with a worse survival in ESRD and HD patients [11, 37]. There were only a few published data in PD patients [11, 38, 39], with limited power due to small sample size. One Korean study [40] advocated the use of Sagittal Abdominal Diameter (SAD) measured by X-ray, and found correlation of central obesity with all-cause and cardiovascular mortality. However, its use is limited by the need of a dedicated X-ray machine and it involves additional radiation exposure, which may not be feasible in a clinic setting. Two small scaled studies by Canadian group [38] with 22 subjects, and Korean group [11] with 84 subjects showed controversial mortality risk from central obesity in PD patients. Another study with a larger sample size from Brazil [39] showed a worse short-term 1-year survival in central obese PD patients. However, the long-term survival could not be assessed due to limited study period. In our study, we recruited 267 subjects, which represented the largest scaled study compared to other studies in same field. Our results reflected neither general nor central obesity predicted 2-year all-cause and cardiovascular disease-related mortality in PD patients.

Central obesity has been reported to have synergistic effect on mortality with physical frailty in HD patients [41]. However, such effect is not yet explored in PD patients. Our study is the first study to prove an additive effect on mortality by physical frailty and central obesity in PD patients. The underlying pathophysiology is probably related to the potentiation of pro-inflammatory state by certain cytokines. Visceral fat in central obese patients secretes numerous pro-inflammatory cytokines including C-reactive protein (CRP), tumor necrosis factor (TNF)-alpha and interleukin (IL)-6 [42, 43]. These cytokines modulate lipid and carbohydrate metabolism [44], and orchestrate the inflammatory pathway by stimulation of T cell proliferation, B cell-related immunoglobulin production and hepatic synthesis of acute phase protein [42]. At the same time, physical frailty is associated with an inflammatory state with high levels of CRP and IL-6. Study has shown elevated CRP and IL-6 levels predict mortality, cardiovascular events and composite outcomes in dialysis patients [42]. Although vascular calcification and arterial stiffening, which could be secondary to chronic inflammation, play a prognostic role on survival of PD patients [45], the cross linkage between frailty, central obesity and mortality appears to be independent from arterial stiffening as reflected by a similar CF-PWV and CR-PWV between frail and non-frail individuals (Table 1, S1 Table), and a lack of correlation between progression of the two parameters with time between frail and non-frail individuals, both in central and non-central obese subgroups. Activation of the inflammatory cascade can also lead to impairment of immune response [46] and subsequent activation of coagulation system causing a higher thrombotic tendency and cardiovascular risk [47]. This may explain why physical frailty had a much higher hazard ratio on all-cause and cardiovascular disease-related mortality in central obese PD patients when compared to all PD patients. It is worthwhile to note that from our result, physical frailty did not cause a worse survival in non-central obese PD patients. The underlying cause remains to be elucidated and further studies should be undertaken in the future to clarify the mechanism.

On top of the proven 1-year mortality risk from frailty in PD patients [5], this study illustrated a longer term mortality risk from frailty in PD patients. However, physical frailty not only confers mortality risk to PD patients, but also imposes a huge burden to the economy and healthcare system as frail individuals are often high users of healthcare resources. In our cohort, frail PD patients were more often required helper assistance in performing dialysis. They also tend to hospitalize more frequent and longer than non-frail individuals, which is in line with other published studies [28, 29, 48]. Notably, frail central obese PD patients required nearly twice as many hospital admissions than non-frail central obese PD patients, and their

total hospital length of stay was also nearly twice in our study. Aside from physical frailty, obesity may also influence clinical outcome in terms of hospitalization in dialysis patients. Patients with higher BMI were associated with a higher incidence of peritonitis-related hospitalization but not in non-peritonitis-related hospitalization [49]. Similarly, central obesity was also identified to be a predictor for cardiovascular events, all-cause hospitalization and mortality [37, 39]. Our data is the first published result both evaluating and comparing the impact of both general and central obesity, as well as identifying the potential interaction with physical frailty on hospitalization parameters in PD patients. In contrast to previous study [37, 39], we identified central obesity as a protective factor associated with a fewer hospital admission and higher survival rate in non-frail PD patients. Similar findings have been reported in patients with stable coronary artery disease patients [50], and heart failure patients especially those with reduced ejection fraction [51, 52]. The proposed mechanisms are related to the extra energy reserve provided by visceral fat [51], and the anti-inflammatory and endotoxin-neutralizing effects mediated by a raised lipoprotein level [52]. In addition, central obese individuals also seek medical attention at the earlier stage of acute illnesses as they tend to be more symptomatic, and their renin-angiotensin-aldosterone system is more attenuated, leading to a higher blood pressure and a better tolerance to cardioprotective medications [52]. On the other hand, general obesity in non-frail PD patients were hospitalized more frequently with a longer duration, regardless of the cause of hospitalization. While most of the available literature focused on recognition of frailty and the associated adverse outcomes in PD patients, our result addressed the impact of body composition in non-frail group and this should not be overlooked. Identification of mechanism underlying such difference between general and central obesity is beyond our scope, but this should be evaluated in future study.

Our study has several limitations. Our data collection is retrospective in nature from a single center with a limited number of study subjects, which may lead to the concern on generalisability. We believe our cohort is a good representation of a Chinese PD cohort as our centre is one of the largest tertiary-care and university-affiliated medical centres. It is also part of the public healthcare system which takes care of 94% of dialysis patients in Hong Kong [53]. We also included both incident and prevalent PD patients, which can lead to a few concerns. Firstly, body composition, fat mass and waist circumference can be altered by the long duration of peritoneal dialysis due to exposure of glucose-containing dialysate [10] and that may have prognostic implications too [39]. However, the marker of central obesity we used i.e. WHR takes both waist and hip circumference into account. Whether there is any change with time depends on the proportion of difference in both parameters. Moreover, univariate analysis reflected PD duration did not predict hospitalization (p = 0.364) and mortality (p = 0.210). All other results reported in this study were adjusted with the duration on PD in their corresponding statistical analyses. On the other hand, we also performed subgroup analysis on patients who were on short PD duration of less than 1 year with a shorter exposure time to glucose-containing dialysate. The survival rate trend is the same as the trend in our whole cohort (S1 Fig) though the difference in survival rates of the 4 groups in subgroup analysis did not reach statistical significance. Together with the other insignificant or barely significant results we presented, this can be attributed to the relatively small sample size in our cohort, which may raise the issue regarding the weight that this study can give. Despite so, our results, as the first published data in this aspect, still give valuable information regarding potential interaction between central obesity, frailty and their association with adverse outcomes in PD patients. Base on this, we suggest future research with a larger sample size to identify any significant interaction and association with the outcomes that we investigated. Secondly, there will also be potential lead-time bias and selection bias favoring surviving patients. However, this can be strength of our study as it can be applied more universally. Moreover, the duration

of follow-up may not be long enough to detect the effect of physical frailty and body composition in survival. Furthermore, our measurement of frailty and body composition is a one-off measurement performed at the recruitment period. These results may also be transiently affected by acute illnesses. Serial measurements of these parameters may provide a better assessment in corresponding aspects. The cross-sectional nature of our data can only allow us to establish association rather than causal relationship. Although we have sufficient power to identify the independent association between two factors, these results do not have sufficient data to analyses the aspect of effect modification.

In summary, we found that physical frailty is prevalent in Chinese PD patients. Frail PD patients were usually older and they often had higher comorbidity loads and poorer nutritional state. They are also more central obese. Physical frailty was an independent predictor for adverse outcomes in PD patients, in terms of a higher hospitalization rate and longer duration of hospitalization, as well as a worse survival with higher all-cause and cardiovascular mortality rates. In subgroup analysis, central and general obesity caused a differential effect on hospitalization parameters in non-frail PD patients. Furthermore, central obesity conferred a numerically but statistically non-significant inferior patient's survival and cardiovascular survival in frail PD patients. On the contrary, central obesity conferred a significantly superior cardiovascular survival in non-frail PD patients. We suggest routine screening of physical frailty and central obesity in PD patients, so as to achieve risk stratification by identification of the high-risk group. Further prospective study is urgently needed to explore on potential interventional measures that can be done to these high-risk individuals to improve their clinical outcomes and survival.

## Supporting information

**S1 Checklist.**
(DOCX)

**S1 Fig. Kaplan-Meier plot of patient survival among patients on peritoneal dialysis for less than 1 year according to the frailty state and central obesity.**
(TIF)

**S1 Table. Baseline clinical and demographic data, statistical tests results.**
(DOCX)

**S2 Table. Relation between hospitalization and general versus central obesity in frail patients.**
(DOCX)

## Author Contributions

**Data curation:** Jack Kit-Chung N. G., Kai-Ming Chow, Vickie Wai-Ki Kwong, Wing-Fai Pang.

**Formal analysis:** Gordon Chun-Kau Chan, Cheuk Chun Szeto.

**Funding acquisition:** Cheuk Chun Szeto.

**Investigation:** Jack Kit-Chung N. G., Vickie Wai-Ki Kwong, Wing-Fai Pang, Phyllis Mei-Shan Cheng, Man-Ching Law, Chi-Bon Leung.

**Methodology:** Vickie Wai-Ki Kwong, Wing-Fai Pang, Phyllis Mei-Shan Cheng, Man-Ching Law, Philip Kam-Tao L. I.

**Project administration:** Cheuk Chun Szeto.

**Resources:** Kai-Ming Chow.

**Supervision:** Kai-Ming Chow, Philip Kam-Tao L. I.

**Validation:** Chi-Bon Leung.

**Writing – original draft:** Gordon Chun-Kau Chan.

**Writing – review & editing:** Cheuk Chun Szeto.

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
