## [Decision Letter · Decision Letter 0]

27 Jul 2020

PONE-D-20-19119

Interaction Between Central Obesity and Frailty on the Clinical Outcome of Peritoneal Dialysis Patients

PLOS ONE

Dear Dr. Szeto,

Thank you for submitting your manuscript to PLOS ONE. After careful consideration, we feel that it has merit but does not fully meet PLOS ONE’s publication criteria as it currently stands. Therefore, we invite you to submit a revised version of the manuscript that addresses the points raised during the review process.

This single-centre retrospective study examined the interaction between central obesity and frailty on outcome in a population of peritoneal dialysis patients. The topic is interesting, but a lot of work will need to be done in order to make this paper eligble for publication in PLOS ONE, see notes by the expert reviewers as well. The methodology is somewhat weak, and authors should consider to only include incident patients rather than incident and prevalent patients. The WHO classification for Obesity should be followed in the analyses, much more detail provided regarding the dialysis treatment and type of PD and medications. The paper should be rewritten according to STROBE guidelines. To improve readability, please remove irrelevant details and p values from the text (consider a table if you deem this is important), and make sure to deliver a focused and strong message. References should be updated as much as possible to include more recent relevant literature. I am willing to give the paper the benefit of the doubt and ask you to make revisions as per above and per reviewers' comments. There is no guarantee that a revised version of the paper would be accepted.

We look forward to receiving your revised manuscript.

Kind regards,

Frank JMF Dor, M.D., Ph.D., FEBS, FRCS

Academic Editor

PLOS ONE

Journal Requirements:

2. Please refer to any post-hoc corrections to correct for multiple comparisons during your statistical analyses. If these were not performed please justify the reasons. Please refer to our statistical reporting guidelines for assistance (https://journals.plos.org/plosone/s/submission-guidelines.#loc-statistical-reporting).

Reviewers' comments:

Reviewer's Responses to Questions

**Comments to the Author**

1. Is the manuscript technically sound, and do the data support the conclusions?

Reviewer #1: Yes

Reviewer #2: Partly

Reviewer #3: No

2. Has the statistical analysis been performed appropriately and rigorously? 

Reviewer #1: Yes

Reviewer #2: No

Reviewer #3: Yes

3. Have the authors made all data underlying the findings in their manuscript fully available?

Reviewer #1: Yes

Reviewer #2: Yes

Reviewer #3: No

4. Is the manuscript presented in an intelligible fashion and written in standard English?

Reviewer #1: Yes

Reviewer #2: No

Reviewer #3: Yes

5. Review Comments to the Author

Reviewer #1: This paper makes an important observation of interaction between central obesity and frailty on outcome in a population of peritoneal dialysis patients.

comments

The paper should be presented according to STROBE guidelines for cohort studies - and in particular there should be a detailed STROBE diagram provided.

There is limited detail on the timing of the interventions with respect to peritoneal dialysis cycles - ie was abdominal girth measured drained out of PD fluid? When were and how were the interventions conducted - what was the accuracy, repeatability or the tests - who conducted them etc? Were all tests conducted at baseline?

Dialysis treatment data is not included in the analysis - for example prescription data (requirement for hypertonic exchanges) or peritoneal transport status. So essentially the dialysis treatment patients received is being ignored in the analysis. The same goes for hypertensive medications.

The study would be much more powerful if it had been conducted on incident rather than prevalent patients.

Reviewer #2: This was a difficult paper to read and follow. There is a huge amount of data with far too many p values for the number of patients and it was not clear what the 'message' of the paper was meant to be. It is already well recognised that frailty is found in about 50% of most dialysis populations, that it is found in all age groups and that frailty is related to poorer outcomes. The text is difficult as it is full of nonsignificant small differences. Furthermore it is not clear why some things were measured like pulse velocity. Another difficulty was the use of abbreviations - far too many - so one had to keep going back to earlier parts of the paper to find out what they stood for.

There are also problems with generasibility of findings as this is a single centre retrospective study with incident and prevalent patients in Chinese patients. As an example, BMI of >25 would not be defined as obesity in many parts of the world,

Reviewer #3: In the paper entitled “Interaction Between Central Obesity and Frailty on the Clinical Outcome of Peritoneal Dialysis Patients” Szeto et. al. has examined the relationship between overall survival (overall and cardiovascular mortality) and central obesity and frailty in patients on peritoneal dialysis (PD). The parameters used to collect the data are relevant. However, the following are the questions, which I would like the authors to address to fill the gap present in the paper.

1. How long were the patients on PD prior to recruiting them on the study? It has been shown in the previous study that patients on PD do gain wait with time and this is associated with changes in the metabolic profile, increased mortality rate and a higher PD failure rate independent of baseline obesity and fluid status. How do you address this issue in your study?

Ref: Kim JK, Park HC, Song YR, Kim HJ, Moon SJ, Kim SG. Effects of Excessive Body Fat Accumulation on Long-Term Outcomes During Peritoneal Dialysis. Perit Dial Int. 2019 20 May-Jun;39(3):268-275. doi: 10.3747/pdi.2018.00164. Epub 2019 Mar 6.

2. In this study, the Waist/Hip (W/H) ratio was measured once at the time of recruitment. Would the reported outcome have changed if serial measurements were carried out at 6-monthly intervals as the W/H ratio does change with the duration of patients on PD? This needs to be included in the discussion section.

Ref: Waist circumference as a predictor of mortality in peritoneal dialysis patients: a follow-up study of 48 months Br J Nutr. 2017 May;117(9):1299-1303.

3. How many patients in the study group were on CAPD, APD and Assisted PD?

4. In the study, obesity has been defined as a BMI over 25. Would further stratification of BMI (<25; 25- 35; and >35) and subgroup analysis have shown difference in the outcomes?

5. What was the incidence of peritonitis and mortality from sepsis in all 4 groups?

Ref: a. The Relationship Between Body Mass Index and Organism-Specific Peritonitis. Perit Dial Int. May-Jun 2018;38(3):206-214.

b. Impact of Obesity on Modality Longevity, Residual Kidney Function, Peritonitis, and Survival Among Incident Peritoneal Dialysis Patients. Am J Kidney Dis. 2018 Jun;71(6):802-813.

6. Relationship between depression, PD and long-term outcomes previously reported by the same authors have not been discussed in the paper?

7. The role of assisted PD in the management of frail patients is being increasingly recognised and there are several publications currently available, which should be included in the paper.

8. References: Majority of citations are more than 10 years old. There are several relevant publications over last 2 years which have not been included and needs attention.

6. PLOS authors have the option to publish the peer review history of their article (what does this mean?). If published, this will include your full peer review and any attached files.

Reviewer #1: **Yes: **Martin Wilkie

Reviewer #2: No

Reviewer #3: **Yes: **Badri Shrestha MD FRCS

---

## [Author Response · Author response to Decision Letter 0]

12 Aug 2020

3 August 2020

Editor in Chief

PLoS One

Dear Editor in Chief,

Re: ‘Interaction Between Central Obesity and Frailty on the Clinical Outcome of Peritoneal Dialysis Patients’ (PONE-D-20-19119)

Thanks for your letter on 27 July 2020. The following revisions have been made according to your recommendations:

Reply to Reviewer 1

1. The paper should be presented according to STROBE guidelines for cohort studies - and in particular there should be a detailed STROBE diagram provided.

The STROBE checklist is completed. A flow diagram on subjects’ inclusion and outcome is added to the manuscript as Fig 1.

2. There is limited detail on the timing of the interventions with respect to peritoneal dialysis cycles - ie was abdominal girth measured drained out of PD fluid? When were and how were the interventions conducted - what was the accuracy, repeatability or the tests - who conducted them etc? Were all tests conducted at baseline?

Page 5, paragraph 4, line 1-2: The anthropometric measurements were performed before infusion of dialysate fluid. 

Page 5, paragraph 2, line 4-7: The waist and hip circumference were measured at the approximate midpoint between the lower margin of the last palpable rib and the top of the iliac crest, and around the widest portion of the buttocks respectively, according to the WHO 2008 report. 

Page 5, paragraph 4, line 1-2: All measurements were performed by the same trained research nurse at baseline. These details are now added to the “Anthropometric and body composition assessment” part of the “Patients and Method” session.

3. Dialysis treatment data is not included in the analysis - for example prescription data (requirement for hypertonic exchanges) or peritoneal transport status. So essentially the dialysis treatment patients received is being ignored in the analysis. The same goes for hypertensive medications.

Page 8, Fig 1; page 33-34, S1 Table: The details for dialysis treatment (in terms of peritoneal dialysis types, and use of icodextrin dialysate), peritoneal transport status (reflected by 4-hour dialysate/plasma creatinine level) and details of medications are added to Table 1 and S1 Table. None of our subject used hypertonic dialysate fluid (i.e. 4.25% dextrose solution) regularly in their PD regimen. 

4. The study would be much more powerful if it had been conducted on incident rather than prevalent patients. 

Please refer to the reply to Point 1 of Editor comment.

Reply to Reviewer 2

1. This was a difficult paper to read and follow. There is a huge amount of data with far too many p values for the number of patients and it was not clear what the 'message' of the paper was meant to be. 

We have modified our manuscript according to the STROBE guideline with appropriate headings and sessions. We have also added new flow diagram, rearranged and simplified our tables to allow readers understand the flow and data of our study.

We have also removed the p values from Table 1, and readers can refer to S1 Table for the corresponding p-value of the data in Table 1. We hope these measures improve readability of our manuscript. 

2. It is already well recognised that frailty is found in about 50% of most dialysis populations, that it is found in all age groups and that frailty is related to poorer outcomes. The text is difficult as it is full of nonsignificant small differences.

There are a few published articles on prevalence and outcome of frail dialysis patients. However, only small number of them covered peritoneal dialysis patients. On the other hand, PD is often regarded as a better dialysis option over HD in elderlies, whom frailty is prevalent in this age group. We hope to provide more information, especially the adverse outcomes and interaction with body composition in the field of frailty in PD. 

3. Furthermore it is not clear why some things were measured like pulse velocity.

Page 18, paragraph 1, line 11-17. Measurement of pulse wave velocity is important for us to understand the possible mechanism between frailty, central obesity and mortality in PD patients. This point is elaborated in the Discussion part.

4. Another difficulty was the use of abbreviations - far too many - so one had to keep going back to earlier parts of the paper to find out what they stood for.

We have minimized the use of abbreviations. To improve readability, the full terminology of abbreviation is labelled below tables and figures.

5. There are also problems with generasibility of findings as this is a single centre retrospective study with incident and prevalent patients in Chinese patients. As an example, BMI of >25 would not be defined as obesity in many parts of the world.

Page 19, paragraph 2, line 2-3: We believe our sample is a good representative of PD patients. Our rationale is elaborated in the Discussion part. 

Page 5, paragraph 2, line 2-3: As for the issue of BMI, we have modified our definition of general obesity as BMI more than or equals 30 kg/m2 according to the WHO definitions. The results were modified according to the new definition.

Reply to Reviewer 3

1. How long were the patients on PD prior to recruiting them on the study? It has been shown in the previous study that patients on PD do gain wait with time and this is associated with changes in the metabolic profile, increased mortality rate and a higher PD failure rate independent of baseline obesity and fluid status. How do you address this issue in your study?

The mean duration on peritoneal dialysis in our cohort was 47.7 ± 51.9 months. 

For the second question, please refer to the reply to Point 1 of Editor comment.

2. In this study, the Waist/Hip (W/H) ratio was measured once at the time of recruitment. Would the reported outcome have changed if serial measurements were carried out at 6-monthly intervals as the W/H ratio does change with the duration of patients on PD? This needs to be included in the discussion section.

Page 19, paragraph 2, line 6-9: It has been reported in the literature that BMI, body fat mass and waist circumference increases with duration of peritoneal dialysis, and such changes have important prognostic implications. However, the change in WHR depends on the degree and proportion of change in waist and hip circumference. Whether WHR changes with time on peritoneal dialysis, together with its prognostic implication are uncertain and should be explored in future study.

3. How many patients in the study group were on CAPD, APD and Assisted PD?

Page 8, Table 1: In our cohort, 233 (87.3%) patients were on continuous ambulatory peritoneal dialysis (CAPD). In the remaining subjects, 6 (2.2% of total) patients were on continuous cyclic peritoneal dialysis (CCPD) and 28 (10.5% of total) patients were on nocturnal intermittent peritoneal dialysis (NIPD).

On the other hand, 32 patients (80%) were on helper-assisted PD in the whole cohort. These details are now added in Table 1 and S1 Table.

4. In the study, obesity has been defined as a BMI over 25. Would further stratification of BMI (<25; 25- 35; and >35) and subgroup analysis have shown difference in the outcomes?

In short, we found no difference in after stratification of BMI and subgroup analysis.

There was no difference between patient survival and cardiovascular survival rates among individuals with BMI <25, 25-35 and >35 (survival rate: 73.5%, 64.8%, 85.7% respectively, log-rank test, p = 0.235) (cardiovascular survival rate: 78.0%, 71.4%, 57.1% respectively, log-rank test, p = 0.269). 

In univariate Cox regression analysis, BMI 25-35 and BMI >35 did not predict patient survival (BMI 25-35: hazard ratio [HR] 1.324, 95% confidence interval [CI] 0.791 - 2.215, p = 0.286; BMI >35: HR 1.955, 95% CI 0.595 - 6.420, p = 0.269) and cardiovascular survival (BMI 25-35: HR 1.266, 95% CI 0.760 - 2.109, p = 0.366; BMI >35: HR 2.295, 95% CI 0.699 - 7.534, p = 0.171) with reference to BMI <25.

Among frail individuals, there were again no difference among individuals with BMI <25, 25-35 and >35 (survival rate: 60.4%, 56.3%, 100% respectively, log-rank test, p = 0.184) (cardiovascular survival rate: 66.0%, 66.7%, 66.7% respectively, log-rank test, p = 0.905). 

In univariate Cox regression analysis, BMI 25-35 did not predict patient survival (HR 1.130, 95% CI 0.617 - 2.070, p = 0.692), while BMI 25-35 and BMI >35 did not predict cardiovascular survival (BMI 25-35: HR 1.017, 95% CI 0.518 - 1.995, p = 0.962; BMI >35: HR 1.357, 95% CI 0.315 - 5.856, p = 0.682) with reference to BMI <25.

Among non-frail individuals, there were only 1 subject with BMI >35. There were again no difference among individuals with BMI <25, 25-35 and >35 (survival rate: 82.3%, 71.9%, 0% respectively, log-rank test, p = 0.070) (cardiovascular survival rate: 86.1%, 75.4%, 0% respectively, log-rank test, p = 0.191). 

In univariate Cox regression analysis, both BMI 25-35 and BMI >35 did not predict patient survival (BMI 25-35: HR 1.663, 95% CI 0.811 - 3.407, p = 0.165; BMI >35: HR 6.834, 95% CI 0.888 - 52.616, p = 0.065) and cardiovascular survival (BMI 25-35: HR 1.635, 95% CI 0.742 - 3.603, p = 0.222; BMI >35: HR 4.265, 95% CI 0.550 - 33.079, p = 0.165) when compared to BMI <25. 

5. What was the incidence of peritonitis and mortality from sepsis in all 4 groups?

There were 77 peritonitis episodes (42 in non-frail group, 35 in frail group). 229 subjects (85.8%) were peritonitis-free. The number of peritonitis-free in non-frail non-central obese, frail non-central obese, non-frail central obese and frail central obese individuals were 68 (87.2%), 45 (93.8%), 56 (81.2%) and 60 (84.7%) respectively. 

After 2 years of observation, 9 subjects (3.4%) died primarily because of infection. The infection-associated mortality rates in non-frail non-central obese, frail non-central obese, non-frail central obese and frail central obese individuals were 3.8%, 2.1%, 4.3% and 2.8% respectively (Log-rank test, p = 0.419).

6. Relationship between depression, PD and long-term outcomes previously reported by the same authors have not been discussed in the paper?

Page 11, paragraph 1, line 1-2: We added the reference of our previously reported depression and frailty in PD patients. However, the theme of this manuscript is mainly about the interaction between frailty, central obesity and mortality. Therefore, we do not discuss about depression in order to make the paper more focused.

7. The role of assisted PD in the management of frail patients is being increasingly recognised and there are several publications currently available, which should be included in the paper.

Page 16, paragraph 2, line 6-8: Recent publications on the role of assisted PD in frail patients were added to the Discussion Part.

8. References: Majority of citations are more than 10 years old. There are several relevant publications over last 2 years which have not been included and needs attention.

Please refer to the reply to Point 4 of Editor comment.

Reply to Editor Comments

Authors should consider to only include incident patients rather than incident and prevalent patients.

We agree with your suggestion that inclusion of incident patients only will reduce potential lead-time and survivor bias, and increase the power of our study. In view of that, we performed adjustments and subgroup analyses to address this issue. 

Page 12, paragraph 2, line 1-7: Firstly, we performed a subgroup survival analysis on patients who were on PD therapy for less than 1 year. In short, the survival rates in non-frail central obese, non-frail non-central obese, frail non-central obese, and non-frail non-central obese individuals were 100%, 82.4%, 76.9% and 61.5% (log-rank test, p = 0.069), and the survival rates were grossly decreasing trend. Although the difference did not reach significance, which is likely contributed to a much smaller sample size after excluding patients on long duration of PD, it still gives us the impression that frailty and central obesity may have potential interaction in mortality in this group of patients. This observation should be confirmed with a study with larger sample size.

Page 20, paragraph 1, line 7-9: Secondly, our analysis reflected PD duration did not predict hospitalisation (p = 0.364) and mortality (p = 0.210). All the other reported results were adjusted with patients’ duration on PD.

Page 20, paragraph 1, line 17: Moreover, inclusion of prevalent PD cases could be a strength of our study as it can be applied universally to all PD cases that we encounter, regardless of their time on PD. 

These details are now added to the Discussion Part. 

The WHO classification for Obesity should be followed in the analyses, much more detail provided regarding the dialysis treatment and type of PD and medications.

The definition of obesity from the WHO classification is adopted in the analyses. The details of analyses with general obesity are modified with the adjusted definition. The details of dialysis treatment and medications are now included in Table 1.

The paper should be rewritten according to STROBE guidelines.

The paper is modified according to the STROBE guideline with a figure showing the flow of our study (Fig 1). The STROBE checklist is also completed.

To improve readability, please remove irrelevant details and p values from the text (consider a table if you deem this is important), and make sure to deliver a focused and strong message. References should be updated as much as possible to include more recent relevant literature.

We have modified our manuscript according to the STROBE guideline with appropriate headings and sessions. New flow diagram is also added in order to allow readers understand the flow of our study. We have also rearranged and simplified our tables. We hope these measures improve readability of our manuscript. The references are also updated with more recent relevant literature replacing the citations that are published more than 10 years ago. There are now only 4 citations that are more than 10 years old left – 2 of them are the landmark studies in frailty and frailty in dialysis (Reference 1 and 2). 

We would like to resubmit the manuscript, with the changes highlighted in yellow, for consideration of publication in the PLoS One. Thank you for reviewing our article and we look forward to hearing your favorable reply.

Yours sincerely,

Dr. Gordon CK Chan

for Drs. JKC Ng, KM Chow, BCH Kwan, VWK Kwong, WF Pang, PMS Cheng, MC Law, CB Leung, PKT Li and CC Szeto

---

## [Decision Letter · Decision Letter 1]

21 Sep 2020

PONE-D-20-19119R1

Interaction Between Central Obesity and Frailty on the Clinical Outcome of Peritoneal Dialysis Patients

PLOS ONE

Dear Dr. Szeto,

Thank you for submitting your manuscript to PLOS ONE. After careful consideration, we feel that it has merit but does not fully meet PLOS ONE’s publication criteria as it currently stands. Therefore, we invite you to submit a revised version of the manuscript that addresses the points raised during the review process.

ACADEMIC EDITOR:

Most of the concerns raised by the reviewers have been addressed by the authors, but i would be keen that the authors address the comments by reviewer 2 in a bit more thorough way and add it to the discussion section:

-can you come with a more crisp and clear clinical message and comment on how much weight can be given to a study in small numbers of patients when outcome differences are barely or not significant.

-There is no explanation why higher Waist:Hip ratio predicts CV events in general population but is supposedly protective in non-frail PD population.

We look forward to receiving your revised manuscript.

Kind regards,

Frank JMF Dor, M.D., Ph.D., FEBS, FRCS

Academic Editor

PLOS ONE

Reviewers' comments:

Reviewer's Responses to Questions

**Comments to the Author**

1. If the authors have adequately addressed your comments raised in a previous round of review and you feel that this manuscript is now acceptable for publication, you may indicate that here to bypass the “Comments to the Author” section, enter your conflict of interest statement in the “Confidential to Editor” section, and submit your "Accept" recommendation.

Reviewer #2: All comments have been addressed

Reviewer #3: All comments have been addressed

2. Is the manuscript technically sound, and do the data support the conclusions?

Reviewer #2: Partly

Reviewer #3: Yes

3. Has the statistical analysis been performed appropriately and rigorously? 

Reviewer #2: Yes

Reviewer #3: Yes

4. Have the authors made all data underlying the findings in their manuscript fully available?

Reviewer #2: Yes

Reviewer #3: Yes

5. Is the manuscript presented in an intelligible fashion and written in standard English?

Reviewer #2: Yes

Reviewer #3: Yes

6. Review Comments to the Author

Reviewer #2: Most of my concerns have been addressed - but I remain mystified by the clinical message and how much weight can be given to a study in small numbers of patients when outcome differences are barely or not significant. There is no explanation why higher Waist:Hip ratio predicts CV events in general population but is supposedly protective in non-frail PD population

Reviewer #3: The authors have addressed all queries raised by the reviewers and I am happy for the manuscript to be accepted for publication.

7. PLOS authors have the option to publish the peer review history of their article (what does this mean?). If published, this will include your full peer review and any attached files.

Reviewer #2: No

Reviewer #3: No

---

## [Author Response · Author response to Decision Letter 1]

8 Oct 2020

2 October 2020

Editor in Chief

PLoS One

Dear Editor in Chief,

Re: ‘Interaction Between Central Obesity and Frailty on the Clinical Outcome of Peritoneal Dialysis Patients’ (PONE-D-20-19119)

Thanks for your letter on 21 September 2020. The following revisions have been made according to your recommendations:

Reply to Editor and Reviewer 2

1. Can you come with a more crisp and clear clinical message and comment on how much weight can be given to a study in small numbers of patients when outcome differences are barely or not significant. 

Page 20, paragraph 2, line 16-22: We acknowledged the small number of patients that we recruited in our study is one of our limitations. It is also one of the reasons that may lead to barely significant or insignificant outcome differences. Despite so, our results, which represent the first published data in this aspect, demonstrated the trend of outcome and provided valuable information by suggesting a potential association and interaction with the adverse outcomes. We suggest future research with a larger sample size to look into any association with the outcomes that we investigated. This limitation and recommendation are now included in the Discussion Part.

2. There is no explanation why higher Waist:Hip ratio predicts CV events in general population but is supposedly protective in non-frail PD population. 

Page 19, paragraph 1, line 16-23: Abdominal obesity is also reported to be a protective factor against mortality in patients with stable coronary artery disease patients, and heart failure patients especially those with reduced ejection fraction. The proposed mechanisms are related to the extra energy reserve provided by visceral fat, and the anti-inflammatory and endotoxin-neutralizing effects mediated by a raised lipoprotein level. In addition, central obese individuals also seek medical attention at the earlier stage of acute illnesses as they tend to be more symptomatic, and their renin-angiotensin-aldosterone system is more attenuated, causing a higher blood pressure and better tolerance to cardioprotective medications. These findings and proposed mechanism are now included in the Discussion Part. 

We would like to resubmit the manuscript, with the changes highlighted in yellow, for consideration of publication in the PLoS One. Thank you for reviewing our article and we look forward to hearing your favorable reply.

Yours sincerely,

Dr. Gordon CK Chan

for Drs. JKC Ng, KM Chow, BCH Kwan, VWK Kwong, WF Pang, PMS Cheng, MC Law, CB Leung, PKT Li and CC Szeto

---

## [Editor Report · Decision Letter 2]

12 Oct 2020

Interaction Between Central Obesity and Frailty on the Clinical Outcome of Peritoneal Dialysis Patients

PONE-D-20-19119R2

Dear Dr. Szeto,

We’re pleased to inform you that your manuscript has been judged scientifically suitable for publication and will be formally accepted for publication once it meets all outstanding technical requirements.

Kind regards,

Frank JMF Dor, M.D., Ph.D., FEBS, FRCS

Academic Editor

PLOS ONE

Additional Editor Comments (optional):

Many thanks for making these final changes.
---

## [Editor Report · Acceptance letter]

14 Oct 2020

PONE-D-20-19119R2 

Interaction Between Central Obesity and Frailty on the Clinical Outcome of Peritoneal Dialysis Patients 

Dear Dr. SZETO:

I'm pleased to inform you that your manuscript has been deemed suitable for publication in PLOS ONE. Congratulations! Your manuscript is now with our production department. 

Kind regards, 

on behalf of

Dr. Frank JMF Dor 

Academic Editor

PLOS ONE